# Marine Collagen-Based Antibacterial Film Reinforced with Graphene and Iron Oxide Nanoparticles

**DOI:** 10.3390/ijms24010648

**Published:** 2022-12-30

**Authors:** Johar Amin Ahmed Abdullah, Emre Yemişken, Antonio Guerrero, Alberto Romero

**Affiliations:** 1Departamento de Ingeniería Química, Escuela Politécnica Superior, Universidad de Sevilla, 41011 Seville, Spain; 2Biology and Hydrobiology Department, Faculty of Science, Istanbul University, 41012 Istanbul, Turkey; 3Departamento de Ingeniería Química, Facultad de Química, Universidad de Sevilla, 41012 Seville, Spain

**Keywords:** marine collagen, extraction, films, graphene oxide, iron oxide, antioxidant, antibacterial

## Abstract

It has become more widely available to use biopolymer-based films as alternatives to conventional plastic-based films due to their non-toxic properties, flexibility, and affordability. However, they are limited in application due to deficiencies in their properties. The marine collagen was the specimen for the present study. Thus, the main objective was to reinforce marine collagen-based films with 1.0% (*w*/*w* of the dry polymer weight) of iron oxide nanoparticles (IO-NPs), graphene oxide nanoparticles (GO-NPs), or a combination of both oxides (GO-NPs/IO-NPs) as antibacterial and antioxidant additives to overcome some of the limitations of the film. In this way, the nanoparticles were incorporated into the film-forming solution (2% *w*/*v* in acetic acid, 0.05 M) and processed by casting. Thereafter, the films were dried and analyzed for their physicochemical, mechanical, microstructural, and functional properties. The results show that the effective combination of GO-NPs/IO-NPs enhanced the physicochemical properties by increasing the water contact angle (WCA) of the films from 77.2 to 84.4° and their transparency (*T*) from 0.5 to 5.2. Furthermore, these nanoparticles added antioxidant and antibacterial value to the films, with free radical inhibition of up to 95.8% and 23.8 mm of bacteria growth inhibition (diameter). As a result, both types of nanoparticles are proposed as suitable additives to be incorporated into films and enhance their different properties.

## 1. Introduction

In recent times, nanocomposites based on biopolymer films have been extensively studied for their potential to provide environmentally friendly food packaging with innovative design features [1]. However, although they possess some benefits, they also have some drawbacks, such as their insufficient barrier properties and mechanical properties to resist the passage of oxygen and water vapor.

Generally, films thinner than 1 mm are highly flexible, transparent, adaptable, impermeable, and suitable for wrapping products of a variety of shapes and sizes [2,3,4]. Considering all these advantages, films also have significant potential for many industries, including food packaging. They can prevent auto-oxidation and reduction reactions, as well as microbe interaction, which serves to extend the shelf life of the product [5]. 

Several critical factors have led to an increase in the demand for films during the past few years: the importance of product safety during transportation, which requires more packaging, and the trend in the industry towards more attractive products, using films for both health and aesthetic benefits [6]. As a result, biopolymers such as polysaccharides, proteins, lipids, and polyesters are used to meet the needs of industry and consumers alike [7,8,9]. Consequently, numerous natural or naturally derived biopolymer-based films (e.g., collagen, chitosan, and cellulose) have been developed that are biodegradable without being toxic [10,11]. In addition, commercial collagen is usually obtained from bovine animals, porcine bones, and the remaining parts of mammalian carcasses. As increasing demand and awareness persist, existing methods begin to be insufficient, and problems related to cost, health issues, immune responses, religious beliefs, etc. begin to emerge. Subsequently, marine collagen has been revealed as a solution for health issues, immune responses, and religious beliefs. An elegant and attractive alternative source of collagen derivatives from fish, jellyfish, marine sponges, etc., has been developed and proposed to overcome these demands, and marine collagen has been revealed and proven to be “disease free” [12]. The benefits of collagen are increasingly recognized, and its application in different areas is increasing. Furthermore, collagen has revealed excellent biocompatibility, biodegradability, and weak antigenicity. Such properties and knowledge led to the development of collagen-based biomedical devices, including drug delivery systems, surgical sutures, hemostatic agents, and tissue-engineering applications [13]. Fish-skin collagen-based wound dressings have shown enhanced tissue regeneration and accelerated wound healing in vitro and in vivo compared with mammalian-based collagen [14]. However, it has significant limitations, including those associated with water, water vapor, and oxygen permeability, as well as weak thermal and mechanical resistance. In addition, it is not suitable for use as resistance to infection. On the other hand, marine collagen is also available to manufacture food packaging materials [12,14]. In both applications, this biopolymer needs to be reinforced with antimicrobial and antioxidant additives [15,16]. Frequently, metal-based nanoparticles, such as silver (Ag-NPs), gold (Ag-NPs) and copper (Cu-NPs), are one of the most suitable agents due to their potential antibacterial capacity [17,18]. Moreover, non-metal functional materials, such as fullerenes and graphene oxide, are reported to have excellent antimicrobial properties [19,20,21]. Furthermore, some researchers have demonstrated the toxic effect of these materials on cells, suppressing cellular growth and multiplication and causing cell death depending on concentrations and duration of exposure [22]. In recent studies about alternative antimicrobials, such as improved biopolymers, there have been considerations for these to be used as promising agents for protection against infections [23,24]. In previous studies, graphene oxide and iron oxide-based materials were shown to have strong antimicrobial properties that inhibit bacterial colonization [25,26,27,28]. However, such studies did not provide further information on the synergic combination of magnetic nanomaterials (as metal-based nanoparticles), graphene oxide (as non-metal-based nanosheets or layers), and polymers that can be achieved when these materials are used as composites. Furthermore, these materials were not characterized for their physicochemical, mechanical, or antioxidant properties. This research shows the invigorating characteristics of polymeric composites and provides a good introduction to the investigation of the synergic combination of metals, non-metals, and biopolymers. 

The main objective of this study was to reinforce marine collagen-based films with 1.0% (*w*/*w* of the dry polymer weight) of IO-NPs, GO-NPs, and GO-NPs/IO-NPs as antibacterial and antioxidant additives to overcome some of the limitations of the film. Further, the films were processed by casting and were then analyzed for their physicochemical, mechanical, microstructural, and functional properties. In this study, the antimicrobial and antioxidant functions, structure, and characterization of marine collagen-based polymer composites of magnetite iron oxide and graphene oxide are presented.

## 2. Results & Discussion

### 2.1. Physicochemical Properties

#### 2.1.1. Water Contact Angle (WCA)

The hydrophobic/hydrophilic properties of the film surface were assessed by determining the WCA of the films [29]. Figure 1 shows the WCAs of collagen alone and collagen reinforced with GO-NPs/IO-NPs, IO-NPs, and GO-NPs. The mean values of this variable are presented in Table 1. As was expected, collagen alone showed hydrophilic behavior, with WCA = 77.6 ± 0.5°, which may be due to the configuration of the hydrophilic compounds within the framework [30]. A similar WCA value (78°) of collagen-based films (type I) has been reported elsewhere [31]. The reinforcement with GO-NPs/IO-NPs, IO-NPs, and GO-NPs significantly increased the WCA to 84.2 ± 0.1, 82.1 ± 0.9, and 79.4 ± 0.1, respectively. This was attributed to the hydrophobic nature of GO-NPs and IO-NPs; thus, the hydrophobic nanoparticles contributed to the impairment of the membrane hydrophilicity and increased the WCA [32]. Similar results have been found with the reinforcement of cellulose acetate/chitosan films with silica nanoparticles [33], cellulose acetate with copper oxide nanoparticles [34], and gelatin/cellulose nanofiber films with zinc oxide nanoparticles [35]. However, several factors can contribute to this phenomenon, such as nanoparticle content, concentration, and size, as well as polymer concentration and nature [30,36]. In this way, GO-NPs displayed larger particle sizes than IO-NPs, which would explain why the IO-NPs contributed more to the increase in WCA than GO-NPs. 

#### 2.1.2. Optical Properties

The photographs of the different films are shown in Figure 2, and detailed results for *T*_600_ % and *T* are shown in Table 1. Further, collagen alone showed higher transmittance at 600 nm, with *T*_600_ = 88.2 ± 1.4%, corresponding to *T* = 0.5 ± 0.1. With the reinforcement of GO-NPs/IO-NPs, IO-NPs, and GO-NPs, a significant decrease was observed in *T*_600_ % to 17.6 ± 1.1, 37.2 ± 0.9, and 34.4 ± 0.8, respectively. Furthermore, the addition of the mixture of GO-NPs/IO-NPs produced the least transparent films in comparison with those reinforced with only IO-NPs or GO-NPs, as demonstrated by the *T* index (Table 1). This is due to the increase in solid content within the film chains, which restricts their mobility and occupies the free spaces, thereby blocking light transmission. Several studies with similar results have also been found in the literature [29,37,38,39,40,41].

### 2.2. Mechanical Properties

Table 2 summarizes the mechanical parameters of different films, as shown in Figure 3. The elastic zone of pure collagen was shorter than its plastic zone. This plastic zone was significantly reduced as the GO-NPs and IO-NPs were incorporated separately, whereas an increase was observed with the combination of both oxides (GO-NPs/IO-NPs). Additionally, the separate incorporation of GO-NPs and IO-NPs led to an increase in brittleness, which is due to the increase in the ultimate tensile strength (Ϭ_max_ = 0.8 and 0.7 MPa, respectively) and Young’s modulus (*E* = 0.8 and 0.5 MPa, respectively), along with a decrease in elongation at break from ε_max_ = 1.2 mm/mm (collagen alone) to ε_max_ = 0.8 and 0.9 mm/mm, respectively. This was explained by the immiscibility of the particles, which restricts the extensibility of the films, thereby forming a non-homogeneous network within the film [42]. However, further incorporation of GO-NPs/IO-NPs did not increase Ϭ_max_ but *E*; this is probably due to the increase in solid content, which may stiffen the matrix of the films [37].

Similar results for other polymer-based films reinforced with nanoparticles have been reported elsewhere [37,43,44]. 

### 2.3. Morphological Properties

The surface of polymers can be made smooth or rough, resulting in profound effects on macromolecule adsorption [45]. The surface morphology of various films is shown in Figure 4. Despite the expectation that pure collagen would have a smooth and homogeneous surface, it showed some irregularities. This may be due to the presence of suspended collagen after centrifugation, which means that collagen is not soluble in its entirety. Furthermore, an increase in surface roughness was observed with the incorporation of the different nanoparticles, being rougher in the films reinforced with GO-NPs/IO-NPs and GO-NPs due to the larger size and aggregation formation [29,30,46,47]. 

### 2.4. Functional Properties

#### 2.4.1. Antioxidant Activity

The aforementioned studies have demonstrated that different types of nanoparticles act as antioxidants. The antioxidant properties of different films were evaluated using DPPH free radicals with gallic acid as a positive control, which showed 95% inhibition. 

In the observations in Figure 5, the DPPH inhibition percentages (*IP*%) are shown. For example, the pure collagen-based film showed the lowest *IP*% *=* 71.1%. On the other hand, the presence of GO-NPs/IO-NPs enhanced this inhibition to 95.8%, probably due to the presence of IO-NPs, which increased the inhibition to 91.9%, and the presence of GO-NPs, which increased the inhibition to 75.0%. The higher antioxidant activity of IO-NPs is possibly due to their ability to produce reactive oxygen species (ROS) and enhance their antioxidant capacity [37,48]. Similar results were found in the literature [37,43].

#### 2.4.2. Antibacterial Activity

The use of nanoparticles, graphene oxide, and magnetic iron oxide nanoparticles is extensive in biomedical applications. This is mainly due to their antimicrobial properties, biocompatibility, and bioavailability [49]. The zones of inhibition of *E. coli* and *S. aureus* produced by GO-NPs/IO-NPs, IO-NPs, and GO-NPs suspensions over time are shown in Figure 5 and summarized in Table 3. Accordingly, the inhibition diameters presenting the antimicrobial effect were between 13.2 and 23.8 mm. Furthermore, this ability depended on nanoparticle types and sizes. Thus, the films reinforced with IO-NPs showed better inhibition than the other samples against both *S. aureus* and *E. coli*. This was attributed to different reasons, including the small size and the ability to generate ROS (e.g., –O^2−^ and –OH), which impair mitochondrial function and damage the DNA and proteins of bacteria [2,18,20]. Previous research has supported similar results with *Pseudomonas aeruginosa* (*P. aeruginosa*) [50]. Additionally, an important factor is nanoparticle agglomeration/aggregation, which can affect the antimicrobial ability, resulting in a decrease in the effective specific surface area and reducing the antimicrobial action [51].

Furthermore, GO-NPs show increasing effects on ROS levels at the highest concentration (100 μg/mL) [52]. In addition, they are non-cytotoxic and nonhazardous at concentrations below 100 µg/mL [53] and can be used as an oral treatment for anemia or iron deficiency [54]. In this context, both types of NPs have been proposed as suitable additives to be incorporated into films and enhance their antimicrobial properties. 

## 3. Materials and Methods

### 3.1. Materials

The marine collagen (Type 1) was provided by Tetis Biotech company (Istanbul, TURKIYE). The DPPH (2,2-diphenyl-1-picrylhydrazyl), methanol and gallic acid (C_7_H_6_O_5_) were supplied by Sigma Aldrich (Darmstadt, Germany). The reagents used were all of analytical quality.

Graphene oxide (GO-NPs, 500 nm and 2–5 layers) was provided by the Aerofen company (Istanbul, TURKIYE).

Iron oxide nanoparticles (IO-NPs, 5–10 nm) were obtained from previous studies [43,55,56,57].

### 3.2. Film Processing Method

The films were processed by casting, as described in previous studies [43], with some modifications. A quantity of 2% *w*/*v* of marine collagen was dissolved in acetic acid (0.05 M) and stirred at 50 ± 5 °C for 2 ± 0.5 h at 600 rpm. The film solution was centrifuged at 10,000 rpm for 10 min to gather the supernatant solution. Subsequently, glycerol (2% *w*/*v*) was added. Additionally, a quantity of 1.0% *w*/*w* of GO-NPs, IO-NPs, and a combination of both oxides (GO-NPs/IO-NPs, total 2%) were dispersed separately into a constant volume (42.7 mL) in the film solution using an ultrasound bath for >30 min (Ultrasounds, J.P Selecta, S.A., Barcelona, Spain) at 0.5 kHz frequency and 0.1 kW sonication power. Finally, solutions were transferred onto Teflon plates and dried in an oven at 50 °C for 72 h. The collagen-based film samples were referred to as collagen alone (1), collagen reinforced with GO-NPs/IO-NPs (2), collagen reinforced with IO-NPs (3), and collagen reinforced with GO-NPs (4). The films were prepared, carefully peeled off, and stored for characterization. 

### 3.3. Characterization Technique

#### 3.3.1. Physicochemical Properties

##### Water Contact Angle (WCA)

The water contact angle (WCA) was implemented to evaluate the hydrophobicity of the film surfaces using the sessile drop method. An ≈ 2 µL droplet of distilled water was placed on the surface of the film (≈1 cm^2^, horizontally leveled) using a µ-syringe. The drop images were taken by a high-resolution camera (108 MP ULTRA-CLEAR CAMERA, Mi 10 T Pro Lunar Silver, Haidian District, Beijing, China) monitored for 15 s, and the WCA was determined on both sides using Image-J free software (V. 1.53q; NIH, Bethesda, MD, USA). A minimum of five tests were performed on each sample to ensure reproducibility, discarding images that differed by more than 2° on either side.

##### Optical Properties

The transmittance and transparency of the films were estimated using UV-vis spectroscopy, according to a prior study [42,43,57]. In brief, the films were cut into pieces of 1 × 2 cm^2^, and their transmittance was recorded at 600 nm using a UV-vis spectrophotometer (Model 8451A, Hewlett-Packard Co., USA). The results were presented as transmittance percentage (*T*_600_ %), and the film transparency (*T*) was calculated from Equation (1) [57]:*T* = −(Log *T*_600_/*t*) (1)
where *T*_600_ is a fraction expressing the amount of light transmitted through the film and *t* is the film thickness in mm. 

#### 3.3.2. Mechanical Properties

The films were subjected to a static tensile test according to a slightly modified standard ISO 527–3:2019 [57] to assess their mechanical characteristics. In this test, an increasing axial force was applied to the films at a rate of 10 mm/min until failure, using a mechanical testing machine (MTS Insight 10 Universal Testing Machine, Darmstadt, Germany). This test provides information on the maximum stress (Ϭ_max_), strain at break (ε_max_), and Young’s modulus (*E*).

#### 3.3.3. Morphological Properties

The surface of the film and its morphological properties were examined by scanning electron microscopy (SEM). The SEM images were captured using a Zeiss EVO microscope (Pleasanton, CA, USA) at an acceleration of 5 kV and 2000× magnification [57]. In addition, the film thicknesses were labeled using free Image-J software (V. 1.53q; NIH, Bethesda, MD, USA). 

#### 3.3.4. Functional Properties

##### Antioxidant Activity

The antioxidant properties of the films were evaluated in accordance with previous studies [43], with some modifications. Additionally, the film-forming solution (1 mL) was admixed with 1 mL of DPPH solution (dissolved in methanol, 40 ppm), followed by centrifugation at 5000 rpm for 10 min at 22 °C. In addition, the absorbance was recorded at 517 nm using a UV spectrophotometer. In this test, gallic acid was used as a positive control. Finally, the DPPH inhibition percentage (*IP %*) was obtained from Equation (2).
*IP* % = (A_u_ − A_i_)/A_u_ × 100 (2)
where A_u_ refers to the uninhibited DPPH absorbance (in the absence of film solution) and A_i_ refers to the inhibited DPPH absorbance (in the presence of film solution).

##### Antibacterial Activity

An agar diffusion experiment was conducted to evaluate the antimicrobial activity of the different samples [37,43]. This involved sterilizing cylindrical films (9 mm in diameter) by immersion in 96% (*v*/*v*) ethanol for 2 min. Furthermore, the films were submerged in agar gels that contained *Staphylococcus aureus* (*S. aureus*) and *Escherichia coli* (*E. coli*). The inhibition diameters surrounding the film were determined after 24, 48 and 72 h of incubation at 37 °C to express the antibacterial capacity using Image-J software (V. 1.53q; NIH, Bethesda, MD, USA). 

#### 3.3.5. Statistical Analysis

The individual samples were measured at least three times in this study. The results are summarized using a mean value and standard deviation estimated using IBM SPSS software (IBM Corp, Armonk, NY, USA. Released 2019. IBM SPSS Statistics for Windows, Version 26.0. Armonk, NY, IBM Corp). Furthermore, significant differences were estimated using a one-way analysis of variance (ANOVA) with a 95% confidence interval (*p* < 0.05).

## 4. Conclusions

Marine collagen is an alternative source of collagen to conventional bovine and porcine collagen. For example, collagen peptides derived from marine species have been shown to illustrate biological activities. In addition, a wide range of implementations of marine collagen have now been identified, and more of them are waiting to be discovered in the future. 

In this study, marine collagen was a suitable candidate for film processing. However, some of its shortcomings limit their application, a possible methodology to overcome these limitations is the film reinforcement with particles of different properties, such as iron and graphene oxide particles. Thus, the incorporation of GO-NPs and IO-NPs has shown great potential to improve physicochemical properties. Thus, considerable decreases in the hydrophilicity by about 9% (water contact angle measurement) and the optical properties by about 90% (transmittance at 600 nm) were achieved with the incorporation of both nanoparticles into the natural marine collagen-based films. Furthermore, mechanical parameters, including the maximum stress and Young’s modulus, were both improved by about 75%. On the other hand, the strain at break decreased with the separate incorporation of GO-NPs and IO-NPs by about 33 and 25%, respectively. The combination of both oxides (GO-NPs/IO-NPs) led to an increase in the strain at break of about 8%, which may be due to the overloaded concentration. Furthermore, the reinforcement with these particles enhanced the inhibition of free radicals, with a maximum inhibition of 95.8%. This benefited the antibacterial properties, which were improved with inhibition diameters of 23.8 and 19.0. mm against *E. coli* and *S. aureus*, respectively. The antioxidant and antimicrobial bio-composites prepared in this study present great solutions for preventing autoxidation and fighting against pathogens in many applications. Therefore, the obtained results of these bio-composites can be helpful for future implementation in different industries, including food packaging and medicine.

Nevertheless, further research is needed to assess the nanoparticle dispersion within the films as well as the thermal properties. Moreover, the authors are planning to study the potential immigration of these particles as well as their effect on film biodegradability.

## Figures and Tables

**Figure 1 ijms-24-00648-f001:**
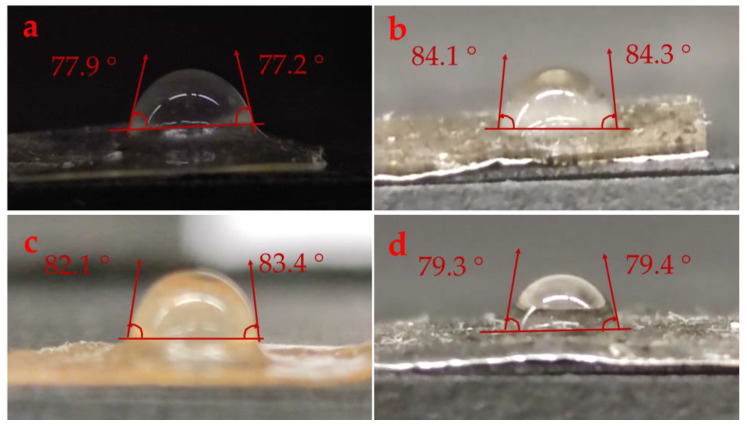
Water contact angle photographs of the different films: (**a**) collagen alone, (**b**) collagen reinforced with GO-NPs/IO-NPs; (**c**) collagen reinforced with IO-NPs; and (**d**) collagen reinforced with GO-NPs.

**Figure 2 ijms-24-00648-f002:**
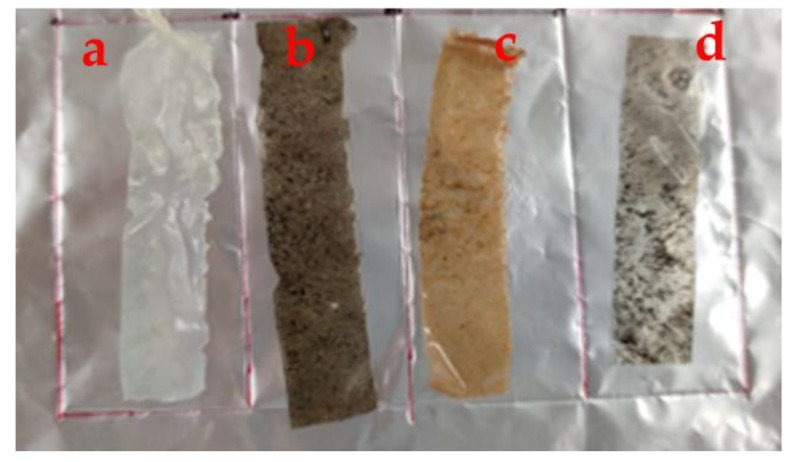
Photographs of the different films: (**a**) collagen alone, (**b**) collagen reinforced with GO-NPs/IO-NPs, (**c**) collagen reinforced with IO-NPs, and (**d**) collagen reinforced with GO-NPs.

**Figure 3 ijms-24-00648-f003:**
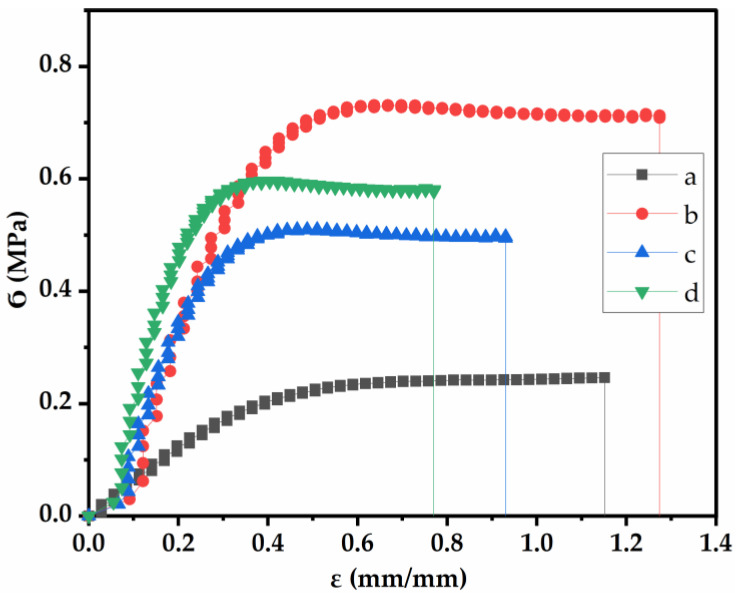
Tensile test profiles of the different films: (a) collagen alone, (b) collagen reinforced with GO-NPs/IO-NPs, (c) collagen reinforced with IO-NPs, and (d) collagen reinforced with GO-NPs.

**Figure 4 ijms-24-00648-f004:**
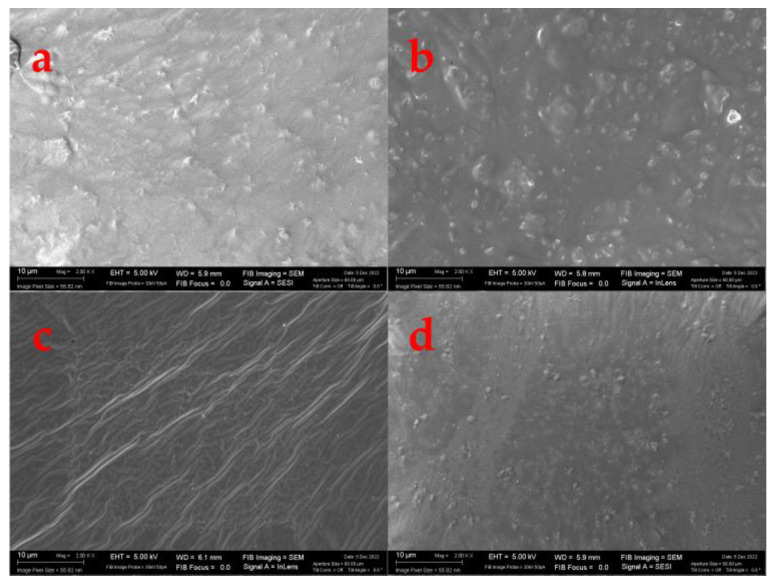
Scanning electron microscopy (SEM) images of the surfaces of the different films: (**a**) collagen alone, (**b**) collagen reinforced with GO-NPs/IO-NPs, (**c**) collagen reinforced with IO-NPs, and (**d**) collagen reinforced with GO-NPs.

**Figure 5 ijms-24-00648-f005:**
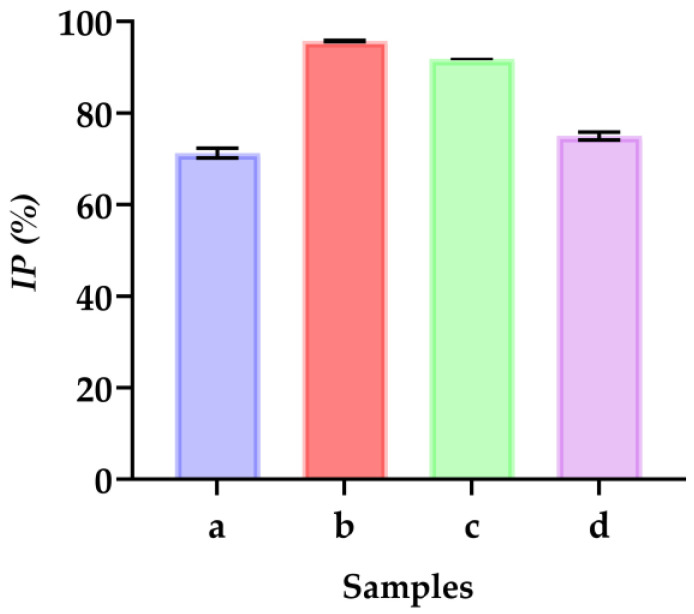
DPPH inhibition percentage of the different films: (a) Collagen alone, (b) Collagen reinforced with GO-NPs/IO-NPs, (c) Collagen reinforced with IO-NPs, and (d) Collagen reinforced with GO-NPs.

**Table 1 ijms-24-00648-t001:** Water contact angle (WCA), transmittance at 600 nm (*T*_600_ %) and transparency (*T*) of the different films: (a) collagen alone, (b) collagen reinforced with GO-NPs/IO-NPs; (c) collagen reinforced with IO-NPs; and (d) collagen reinforced with GO-NPs.

Sample		*WCA* (°)	*T*_600_ (%)	*T*
Commercial Value		96	-	-
a	Collagen alone	77.6 ± 0.5 ^d^	88.2 ± 1.4 ^f^	0.5 ± 0.1 ^c^
b	Collagen reinforced with GO-NPs/IO-NPs	84.2 ± 0.1 ^a^	17.6 ± 1.1 ^e^	5.2 ± 0.2 ^a^
c	Collagen reinforced with IO-NPs	82.1 ± 0.9 ^b^	37.2 ± 0.9 ^d^	3.7 ± 0.1 ^b^
d	Collagen reinforced with GO-NPs	79.4 ± 0.1 ^c^	34.4 ± 0.8 ^c^	3.9 ± 0.1 ^b^

Note: The same superscript letters (^a–f^) in each column indicate homogeneity of variances (*p* < 0.05).

**Table 2 ijms-24-00648-t002:** Results for thicknesses, maximum stress (Ϭ_max_), strain at break (ε_max_), and Young’s modulus (*E*) of the different films: (a). collagen alone, (b). collagen reinforced with GO-NPs/IO-NPs; (c). collagen reinforced with IO-NPs; and (d). collagen reinforced with GO-NPs.

Sample		Thickness (µm)	Ϭ_max_ (MPa)	ε_max_ (mm/mm)	*E* (MPa)
a	Collagen alone	99.9 ± 2.1 ^c^	0.2 ^d^	1.2 ^b^	0.2 ^d^
b	Collagen reinforced with GO-NPs/IO-NPs	136.5 ± 1.3 ^a^	0.7 ^b^	1.3 ^a^	0.6 ^b^
c	Collagen reinforced with IO-NPs	116.1 ± 1.2 ^b^	0.5 ^c^	0.9 ^c^	0.5 ^c^
d	Collagen reinforced with GO-NPs	118.9 ± 1.9 ^b^	0.8 ^a^	0.8 ^d^	0.8 ^a^

Note: The same superscript letters (^a–d^) in each column indicate homogeneity of variances (*p* < 0.05).

**Table 3 ijms-24-00648-t003:** Inhibition diameters (mm) produced by the different films: (a) Collagen alone, (b) Collagen reinforced with GO-NPs/IO-NPs, (c) Collagen reinforced with IO-NPs, and (d) Collagen reinforced with GO-NPs.

		*E. coli*			*S. aureus*		
Sample		0 h	24 h	48 h	0 h	24 h	48 h
a	Collagen	10	-	-	10	-	-
b	Collagen reinforced with GO-NPs/IO-NPs	10	15.0 ± 0.2 ^b^	15.5 ± 0.6 ^b^	10	13.2 ± 0.9 ^b^	13.4 ± 1.0 ^c^
c	Collagen reinforced with IO-NPs	10	23.8 ± 1.1 ^a^	21.5 ± 0.3 ^a^	10	18.7 ± 1.1 ^a^	19.0 ± 1.2 ^a^
d	Collagen reinforced with GO-NPs	10	15.5 ± 0.9 ^b^	14.9 ± 0.3 ^b^	10	14.9 ± 0.2 ^b^	15.0 ± 0.3 ^b^

Note: The same superscript letters (^a–d^) in each column indicate homogeneity of variances (*p* < 0.05).

## Data Availability

The data presented in this study are available on request from the corresponding author.

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
