# Peer review of "Marine Collagen-Based Antibacterial Film Reinforced with Graphene and Iron Oxide Nanoparticles"

_ijms, 2022, doi:10.3390/ijms24010648_

Round 1
Reviewer 1 Report
Authors have reported fabrication and evaluation Marine Collagen-Based Antimicrobial Film Reinforced with Graphene and Iron Oxide Nanoparticles. Work is very well presents but need some major improvements.
1. In introduction the advantages of used material over other antimicrobial materials is important so go through some literature and revise. For that you can refere and cite following relevant article https://doi.org/10.1002/pat.5457
2. State the novelty of present study in comparison with available film formulations
3. In method section author states that they have modified the available film formation process. Kindly state the modifications
4. How the reinforcement of Graphene and Iron oxide was characterized?
5. Include the antibacterial activity study procedure in method section
6. Rectify the minor grammar and sentence errors
Reviewer 2 Report
It is interesting to read the manuscript “Marine Collagen-Based Antimicrobial Film Reinforced with Graphene and Iron Oxide Nanoparticles”. I appreciate the authors' efforts in conducting the study, which will be more beneficial to scholars working in the field. However, if the following changes are made and they are incorporated into the manuscript, it could be considered for publication in International Journal of Molecular Sciences.
1. I advise authors to modify the title as "Marine Collagen-Based Antibacterial Film Reinforced with Graphene and Iron Oxide Nanoparticles”.
2. Line 51-88There is too much information regarding marine collagen. Authors should think about making it concise.
3. Section 2.1. – From the list, remove acetic acid. Since it is a highly common reagent, its inclusion is not required.
4. Line 166 - Staphylococcus aureus (S. au) and Escherichia coli (E. col) to be changed as Staphylococcus aureus (S. aureus) and Escherichia coli (E. coli).
5. Figure 1 – Not clear. High resolution is required.
6. Change Table 1's title. For example
Table 1. Water contact angle of different films.
7. The table's remaining information should be in the footnote rather than next to the table's title. [a-f homogeneity of variances (P < 0.05)]
8. In place of that, mention samples 1, 2, 3, and 4. Mention 1. Collagen alone, 2. Collagen reinforced with GO-NPs/IO- 203 NPs, 3. Collagen reinforced with IO-NPs, and 4. Collagen reinforced with GO-NPs. This details to be within the table.
9. The legend of Figure 2 needs to be revised.
10. As indicated in comments 6, 7, and 8, Table 2 also must be modified.
11. Figure 6 should be deleted because it is unnecessary. Because of the values in Table 3.
12. As indicated in comments 6, 7, and 8, Table 3 also must be modified.
13. The results section has excellent writing. The lack of explanation in the discussion of each segment was another thing I noticed. Instead of giving more background information about the literature, I would advise the authors to refocus their discussion on each part so that it is evident how the research's findings fit into the larger context of what is happening right now with antibacterial film.
14. In the conclusion, I suggest that authors to offer a critical justification on their findings and observations. As a result, everyone will understand the value of this research. Potential points of view must be covered in the conclusion as well. The importance of this study piece should be emphasised by the author.
Reviewer 3 Report
In this article " Marine Collagen-Based Antimicrobial Film Reinforced with Graphene and Iron Oxide Nanoparticles ", author mentioned about the Marine Collagen-Based Graphene and Iron Oxide Nanoparticles with their Antimicrobial assessments. Overall, based the data provided, it can be considered for publication in this journal. However, there are some issues that have to be fixed before publication;
For increasing the interest of readers, the abstract could be more specific, could be divided into four sections: scope, objectives, methods, results, and conclusions. Also, the abstract section should be completed with the study's results in numerical form.
Please specify the novelty, significance, technical merit of the study in a better way.
English is very poorly presented throughout the manuscript. as well as the presentation is also very poor, should be uniform.
Please revised introduction with proper consequences. and could be better if compared with some other applications of the same material like Journal of Environmental Chemical Engineering 9 (5), 106111.
For increasing the interest of readers, the conclusion should also revised with same pattern (based on some numerical based results).
Make a table of comparison of this present work and other similar techniques from previously published a study in terms of operational parameters.
Please check all errors like found in the caption of Fig2.
The applications of findings/results should be clearly stated and mention some valuable inputs.
Some errors regarding the sub/super script, spacing and typo need to consider throughout the manuscript. (specially font type and size of equation 3)
Make sure that the format of references are uniform. Moreover, please add some more references to support this study.
Round 2
Reviewer 1 Report
Accept in present form
Reviewer 2 Report
The author addressed all of my queries and made the necessary changes in the manuscript. I therefore recommend that it to be considered for publication in IJMS in its current form.